# Food Insecurity among International Migrants during the COVID-19 Pandemic: A Scoping Review

**DOI:** 10.3390/ijerph20075273

**Published:** 2023-03-27

**Authors:** Doua Ahmed, Pierina Benavente, Esperanza Diaz

**Affiliations:** 1Centre of International Health, Department of Global Public Health and Primary Care, Faculty of Medicine, University of Bergen, 5020 Bergen, Norway; 2Pandemic Centre, Department of Global Public Health and Primary Care, Faculty of Medicine, University of Bergen, 5020 Bergen, Norway

**Keywords:** scoping review, COVID-19, food insecurity, food security, health, international migrant

## Abstract

The SARS-CoV-2 coronavirus and the measures imposed to control it have impacted food security globally, particularly among vulnerable populations. Food insecurity, in turn, has repercussions on health, exacerbating pre-existing inequalities. This scoping review maps the literature describing associations between the COVID-19 pandemic and food insecurity among migrants, with a particular view toward health. A total of 909 papers were extracted through four electronic databases, and 46 studies were included. The migrant populations described originated mainly from Latin America (11/46) and were located in North America (21/46). Most studies included refugees and asylum seekers (20/46). The main challenges described were financial hardship (28/46), the effect of migrants’ documentation status on using public food aid (13/46), and the suspension of or reduction in humanitarian assistance due to the economic recession (7/46). The impact of food insecurity on migrants’ mental and physical health was described in 26 of the 46 studies. Authorities in all destination countries should focus their attention and efforts into ensuring nutrition security for migrants in a holistic way, including their economic and legal integration, to be better prepared for health crises in the future.

## 1. Introduction

The COVID-19 pandemic has negatively impacted different aspects of life. The virus has killed more than six million people around the world [1], and the measures to control the spread of the virus have affected people’s mental and physical health globally [2,3]. In addition, COVID-19’s containment measures and the resulting global economic recession have had several devastating outcomes [4,5].

Indeed, the pandemic has disproportionately impacted refugees, asylum seekers, and other migrants who live in vulnerable conditions and have a higher risk of being infected and dying from COVID-19 [6]. However, migrants constitute a heterogenic group that has been affected differently, depending on factors such as their country of origin and destination, the reason for migration, and their status of documentation, among others [7,8]. In addition, people in vulnerable situations experienced more financial hardship and other social consequences during the pandemic, which might be attributed to pre-existing socioeconomic inequalities [9]. Poorer socioeconomic conditions can in turn increase vulnerability to infection and disease, creating vicious cycles that are reinforced during a pandemic [10].

Food security is achieved when “all people, at all times have the physical and economic access to sufficient, safe and nutritious food that meet their dietary needs and food preferences for an active and healthy life” [11]. Global efforts are directed to ensure food security for all people, as this is directly connected to the first and second sustainable developmental goals (SDGs), “no poverty” and “zero hunger”, respectively. At the same time, food security affects and is affected by other SDGs such as good health and wellbeing, quality education, and clean water and sanitation [12]. Moreover, food security is a crucial element in maintaining good physical [13,14] and mental health [15,16].

Food security is a complex and multidimensional concept. To reflect upon this, we used a framework developed by Gibson based on the Committee on Food Security and the Food Insecurity and Vulnerability Information and Mapping Systems initiative [17]. This framework divides food security into four different levels (individual, household, national, and underlying conditions). There are several factors related to food security in each (Figure 1), including individual health status [17].

Although often used as the mere opposite of food security, food insecurity is defined as “the inability to acquire or consume an adequate diet quality or sufficient quantity of food in socially acceptable ways, or the uncertainty that one will be able to do so”, which is usually related to economic difficulties and constraints on resources [18]. Although food security has mostly been studied in low- and middle-income countries and in studies conducted by non-governmental organisations (NGOs) and international humanitarian organisations, the concern regarding food insecurity is also growing in high-income countries, specifically among the most deprived parts of the population. According to a Norwegian study, 3% of Norwegians have experienced food insecurity, and 93% of refugees were found to be food insecure [19].

Food security has obtained increasing attention during the COVID-19 pandemic, with a growing number of papers describing the situation for the majority population. These studies acknowledge the complexity of the associations between the pandemic and the key role of sociodemographic factors [20,21]. However, they seldom include migration as an additional factor to consider in these associations. So far, most of the information regarding food security/insecurity among migrants during the pandemic comes from NGOs and international humanitarian organisations working in low- and middle-income countries [22]. Although this information is extremely valuable, humanitarian organisations often have advocacy as one of their roles and do not have the resources to conduct independent investigations available to the research community.

Reliable peer-reviewed evidence of high quality about food insecurity among migrants is necessary for researchers to understand the intricacies of associations between health and migration background and for policymakers to make adequate decisions, specifically during health crises such as the pandemic, when health, economy, and health services are compromised. This study aims to provide an overview of published research on how the coronavirus pandemic has affected food security among migrants. Furthermore, we want to map the existing literature to investigate the extent, range, and characteristics of the evidence related to migrants’ food security amid the COVID-19 pandemic. In light of this information, we try to identify any gaps in knowledge related to the levels and factors of the food insecurity framework, with a special emphasis on health, as well as to different risk factors related to migration background. Even though food insecurity is of utter importance for internally displaced people, too, given that the pandemic posed strict constraints on movements across national borders, for this paper, we study migrants that have moved to a new country.

## 2. Materials and Methods

### 2.1. Study Design

We conducted a scoping review following the methodology proposed by Arksey and O’Malley [23]. According to this approach, the following steps were followed to ensure the rigour and transparency of the method. First, the following research question was identified: What kind of peer-reviewed literature has been published about food security among international migrants during the COVID-19 pandemic? We then searched for relevant studies and carried out a selection process to choose only those papers that met the inclusion criteria. Thereafter, data obtained from the selected studies were charted. Finally, the results were summarised and reported.

To be included in this review, studies had to be written in English, and they had to include three topics: food security/food insecurity, international migrants, and COVID-19. We excluded studies on internal migrants or minorities and ethnic groups, systematic and scoping reviews, and articles that discussed the effect of COVID-19 on agriculture and food production due to the shortage of migrant farmworkers.

### 2.2. Search Strategy

In order to identify articles relevant to this scoping review’s focus and acquire a multidisciplinary approach with regard to how COVID-19 could influence the food security of migrants, our search used four electronic databases: Embase, Medline, Web of Science, and PsychINFO. The search strategy was conducted with the help of a librarian expert from the University of Bergen, particularly to draft search terms. The PICO search strategy tool was used to determine our search terms (Table 1). Medical subject headings (MeSH) (emigrants, immigrants, migrants, transients and refugees, food supply, food insecurity, food security, coronavirus infections, Betacoronavirus, pandemics, and COVID-19), truncated free-text terms (emigra*, immigra*, migrant*, transient*, refugee*, asylum seeker*, ethni*) (coronavirus*, pandemic*, corona virus*, virus disease*, virus infection*), and also food security, food availability, food accessibility, and food utilisation were used in this study. We limited the search until November 2022. The final search was carried out on 26 January 2023, exported to EndNote bibliographic manager, and the duplicates removed.

### 2.3. Data Extraction

In the first search, a total of 909 papers were obtained (323 from Embase, 220 from Medline, 59 from PsycINFO, and 307 from Web of Science). Of those, 292 were duplicates and therefore omitted. Initially, articles were screened based on titles and abstracts; at this point, 524 were excluded, since they did not meet the inclusion criteria. Any doubts over eligibility were resolved through discussion within the research team. If the decision of inclusion was not clear based on the title and abstract, the articles were then selected for the next round of review. A total of 93 studies were selected in the first round and were reviewed through full-text reading. Those studies that did not meet the inclusion criteria were eliminated, and all doubts during this process were discussed within the team. Data were extracted from the selected articles using an Excel template. The variables used in this template were: characteristics of the paper (author, journal, publication year, and the country in which the research was conducted), research design and method, features of the migrant population in the study (country of origin and destination, migrant category in terms of documentation status and/or reasons for migration), objectives, outcomes, and results. Regarding food (in)security, the following variables were also added in the Excel template: tools or methods used to measure/assess food (in)security in the study, level at which food (in)security was discussed (i.e., at the individual, household, or national level), health issues related to food (in)security, and the mechanism by which the spread of COVID-19 and its containment measures impacted the food security of international migrants. These variables were selected based on the objectives and the framework of the study. During the full-text review stage, a new variable regarding the possible effect of migrants’ documentation status on using food aid was added to the template. It was observed that many studies focused on the impact on the agriculture and food production sectors caused by the shortage of migrant farmworkers due to the pandemic. Those studies did not discuss the direct effect of COVID-19 on the food security of the migrants themselves, so they were excluded based on a discussion within the research team, and the exclusion criteria were adjusted.

A total of 46 studies that met the inclusion criteria were included in this scoping review. Figure 2 shows the screening and selection process followed [24].

## 3. Results

Thirty-seven papers contained original research, four papers were commentaries [25,26,27,28], one was a viewpoint [29], one was an opinion piece [30], two were literature searches [31,32], and one was a letter to the editor [33]. (Table 2). Thirty-five studies mainly or totally used the term “food insecurity” [26,27,28,29,31,33,34,35,36,37,38,39,40,41,42,43,44,45,46,47,48,49,50,51,52,53,54,55,56,57,58,59,60,61,62], six used “food security” [25,30,32,63,64,65], and five used both [66,67,68,69,70].

Among the 46 studies included in this scoping review [25,26,27,28,29,30,31,32,33,34,35,36,37,38,39,40,41,42,43,44,45,46,47,48,49,50,51,52,53,54,55,56,57,58,59,60,61,62,63,64,65,66,67,68,69,70], half of them were conducted in America [28,29,31,33,40,42,43,46,47,48,49,50,54,55,57,59,61,63,66,67,68,69,70], 11 studies were conducted in Asia [30,34,36,38,39,41,44,45,52,62,64], and 6 studies were conducted in Africa [25,26,27,32,53,65]. Only four of the studies were conducted in Europe, and two in Oceania [51,56].

Eight studies [36,38,39,41,44,62,64,67] were focused exclusively on migrants from Asia (China, Bangladesh, India, Myanmar, Pakistan, and Syria). Eight papers included exclusively African migrants (from Ghana, Zambia, Congo, Burundi, South Sudan, Somalia, Eritrea, Sudan, and other African countries that were not specified) [25,26,27,32,40,45,65,66]. Eleven studies focused exclusively on Latin American migrants (from El Salvador, Mexico, Guatemala, Venezuela, Cuba, Honduras, Puerto Rico, and other countries in Central and South America) [28,31,33,42,47,48,54,61,63,69,70], and two studies included migrants from Oceania (Republic of the Marshall Islands) [43,49]. Ten studies did not specify the region of origin of the migrant population [30,37,46,50,51,52,53,56,58,59]. Seven studies presented a combination of migrants from different regions (including non-EU European countries) [29,34,35,55,57,60,68] (Figure 3).

With regard to the categories of the migrant population, refugees and asylum seekers were studied in 20 papers [25,26,32,33,37,38,40,41,45,47,52,53,55,57,58,62,63,64,65]. Migrant workers were presented in five papers [30,34,36,57,60], and three papers studied international students [39,56,66]. The sample in seven studies included migrant families [42,46,57,61,67,68,70], and five also included undocumented migrants [26,35,52,59,60].

Food (in)security was included in all papers, as it was a condition for inclusion. However, among the thirty-seven papers containing original research, food (in)security was the main focus in ten studies [39,42,50,51,56,58,62,66,68,69], another ten studies focused on migrants’ health and wellbeing [34,35,41,53,54,56,57,59,61,65], and the impact of COVID-19 on living conditions was the focus in eleven studies [36,37,38,46,48,49,55,60,63,64,67]. Two studies focused on education during the pandemic [43,52], and in one study, the primary outcome was the knowledge and information gathered about COVID-19 [40].

In more than half of the studies selected (28 out of 46), food insecurity among migrants was explained by the financial hardship resulting from the COVID-19 pandemic (loss of jobs, decreased working hours, and low incomes that prevented migrants from purchasing nutritious food or forced them to adapt their diet either by skipping meals, consuming cheap, poor-quality food, and/or prioritising children’s food) [27,28,30,31,32,33,35,36,37,38,40,44,46,48,49,50,52,54,55,56,58,60,61,62,63,65,67,69]. In addition, thirteen of the studies showed that the documentation status of migrants impacted the utilisation of national food assistance [26,27,28,29,30,31,35,42,56,59,60,62,69], and seven studies discussed how the economic recession caused by the pandemic led to minimising the food rations served to refugees and asylum seekers in refugee camps [25,38,58,62,63,64,65].

The spread of the COVID-19 pandemic has negatively impacted the food security of international migrants through other paths. Five studies mentioned the unavailability of certain food items either due to a disturbed supply chain or panic buying as a consequence of the pandemic [28,29,37,66,69]. Moreover, six studies described the role of food safety, which relates to the hygiene and settings in which food is prepared and served, and health consciousness regarding the consumption of unfamiliar or low-quality food [27,28,36,39,58,69]. In addition, the closure of schools, one of the containment measures applied by many governments around the globe, was related to food insecurity in migrant households in six of the studies [27,31,40,52,65,69]. Language was also considered a barrier that may have affected the use of government assistance programmes in two of the articles selected [28,40]. Finally, 3 of the 46 studies did not describe exactly the mechanism through which the COVID-19 pandemic affected the food security of international migrants [34,41,43,46,51,53,57,64,68,70] (Table 3).

Food insecurity was related to health in 26 of the 46 studies. As Table 4 shows, eighteen of the selected studies described the negative effect of food insecurity on the mental health of international migrants [25,26,28,34,36,39,40,41,53,54,56,57,59,60,61,63,65,66]. In particular, food insecurity could be related to anxiety and depression. Eight studies showed a negative impact of food insecurity on physical health [25,27,32,36,37,47,62,68]. Two studies showed the negative impact of food insecurity on both mental and physical health [25,36]. At the same time, eleven studies described the effect that COVID-19’s containment measures and their socioeconomic impact on the mental wellbeing of migrants rather than or in addition to studying the direct effect of food insecurity on health [26,28,35,37,38,43,44,48,49,67,70].

## 4. Discussion

This scoping review gives an overview of the published peer-reviewed literature on the effect of COVID-19 on food insecurity among international migrants. It covers the pandemic period up to November 2022, by which time most countries had implemented and thereafter ended the toughest containment measures.

Migrants’ food insecurity was significantly exacerbated during the COVID-19 pandemic [68,71]. According to our findings, one of the major causes of this was the financial hardship that resulted from the containment measures put in place to hinder the spread of COVID-19 and the way in which these measures precipitated an economic recession [72]. Migrant populations have experienced higher job losses and decreased incomes than the majority populations in the wake of the pandemic, which eventually disrupted their access to adequate nutritious food [9,73,74]. This finding builds upon the existing evidence that shows that migrants experienced food insecurity before the pandemic to a higher degree than non-migrants as a consequence of several socioeconomic inequalities [14]. However, the studies in this scoping review show that worsening food insecurity among migrants since the start of the pandemic could not be explained by pre-COVID-19 risk factors of food insecurity or COVID-19-related income loss alone [42].

In addition to economic factors, migrants’ documentation status was associated with food insecurity during the COVID-19 pandemic. Indeed, several studies in this scoping review showed that undocumented migrants had impaired access to health services as well as to government food assistance. For this group, avoiding public help was also associated with the fear of being legally exposed and then deported to their home countries. Furthermore, migrants with legal residency sometimes preferred not to use food aid because of the regulations in some host countries that prevent migrants from having legal permanent residence if they benefit from certain national assistance. This made migrants avoid other relief programmes that help mitigate the impact of the pandemic even if they were eligible for them [75,76]. Thus, the legal determinants of health seem to also apply to further endangering the food security of those migrants who are most vulnerable [77].

As shown in some studies, the closure of schools during the pandemic worsened the situation of food insecurity among migrant households. Before the pandemic, migrant children used to have cooked meals in their schools, which helped them to maintain their minimum nutritional requirements. However, during the COVID-19 pandemic, children stayed at home and lost out on school meals, increasing the food insecurity burden for migrant schoolchildren and their families [78]. These results are consistent with a study in the United States that reports that more than 1.15 billion school meals were missed between March and May 2020 alone [79]. In addition, a considerable number of studies have discussed the effect of other containment measures, such as closed borders and travel bans, in producing a scarcity of migrant workers on farms and in food production and the ways in which this precipitated global food insecurity [80,81]. However, such studies have been excluded from this scoping review, as they did not describe the direct effect of food insecurity on migrants themselves.

Food insecurity and mental health problems can reinforce each other [82,83]. Almost a third of the studies in this scoping review focused on migrants’ psychological disorders that emerged as a consequence of experiencing food insecurity during the pandemic. These studies revealed a range of symptoms, from sadness and psychological distress to depression and anxiety. This is in line with Nagata et al., who illustrated how food insecurity during the COVID-19 pandemic is associated with poor mental health [84]. At the same time, evidence showed mental health deterioration during the COVID-19 lockdown because of social distancing, reduced social support, financial insecurity, and worries related to health and COVID-19 infection [71,85,86].

Some of the papers in this scoping review reported on the impact the pandemic has had on the physical health of migrants because of the lack of a balanced and nutritious diet. The literature shows that food-insecure people have a higher risk of infections due to low immunity [87]. In addition, Gowda et al. found that over time, food insecurity was associated with two forms of malnutrition: obesity and undernutrition [88]. Moreover, other studies have shown that children living in food-insecure households have about three times higher odds of developing iron deficiency anaemia than those residing in food-secure households and are frequently prone to infections that hinder their normal growth and development [89,90]. Although we have not found literature on food insecurity and health specific to migrants before the COVID-19 pandemic, we can infer that if food insecurity among migrants increases, their health problems will also increase.

It is noteworthy that of the several hundred thousand peer-reviewed papers related to COVID-19 published during the pandemic, only a few articles discussed how COVID-19 impacted the food security of international migrants. Furthermore, only four of the studies in this scoping review were carried out in European countries, revealing a lack of research awareness on this subject and a disregard of the fact that food insecurity among migrants is also increasing in high-income countries such as Norway [91]. Based on the food security framework used in this study, we have found several gaps in the research and in knowledge. First, only two studies focused on national/regional levels and on contextual/underlying conditions. Second, few studies connected food security and the health status of migrants. Last, food security was not always the primary outcome or explanatory factor and was measured with different instruments, often non-comparable and poorly described. Addressing these gaps would help to identify factors at all levels and to understand their connections. This is crucial in order to make comparisons and implement policies and interventions.

With regard to the recommendations in the field of research on migration and health [92], although definitions and classifications of migrant groups were incomplete or omitted in some of the studies selected, most of the migrants presented in the studies included in this scoping review belong to groups in very vulnerable situations: refugees and asylum seekers, undocumented migrants, and precarious migrant workers. Based on the existing evidence, different migrant workers, not only those in precarious situations, have experienced the loss of jobs and/or decreasing wages affecting their health and living conditions [3,93]. More focus is needed on the impact that COVID-19 has had on the health and food security status of different types of international migrant workers in different countries.

This scoping review has some limitations. First, because it includes only studies written in English, it is possible that many relevant national and international studies in other languages have been missed. Further, papers that do not describe the direct association between the pandemic and food insecurity with regard to migrants themselves were excluded, which led us to ignore valuable information on how COVID-19 disturbed general food supply chains and the food production process, which exacerbated food insecurity in general. The papers obtained in our search were heterogeneous in terms of definitions (often using food security and insecurity interchangeably) and design and method, which made creating comparisons and finding common topics challenging. However, four different databases were used for the search, making our scoping review comprehensive and multidisciplinary. Lastly, by setting our results in light of a food security framework and the recommendations for research in the research field of migrant health, we have suggested some gaps in the literature in terms of the volume of literature, migrant-related factors that are covered, and the description and instrumentalisation of food insecurity as the main outcome or explaining factor. However, a scoping review gives some indications of the research published but does not cover all literature. Nevertheless, we acknowledge our scoping review’s limitations in detecting all literature and appreciate that other ways of looking for gaps might be more relevant.

## 5. Conclusions

Migrants have been overrepresented in COVID-19 statistics, and the spread of the pandemic has also vastly increased food insecurity among international migrants through financial, legal, and other pathways. Food insecurity has probably further worsened migrants’ physical and mental health, exacerbating pre-existing inequities in health. However, very few scientific papers describe this situation, and most studies have been conducted on only two continents and included migrants from few countries of origin. Therefore, authorities in all destination countries should put more attention and effort into ensuring nutrition security for different types of migrants in a holistic way, including their economic and legal integration, to be better prepared to contain health crises and epidemics in the future.

## Figures and Tables

**Figure 1 ijerph-20-05273-f001:**
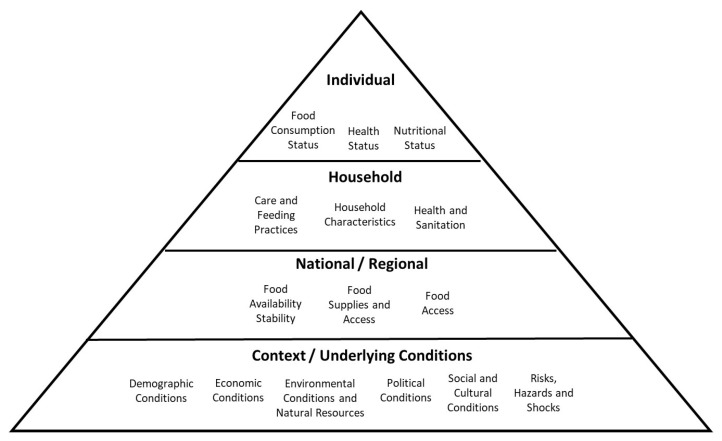
Levels of food security adapted from [17].

**Figure 2 ijerph-20-05273-f002:**
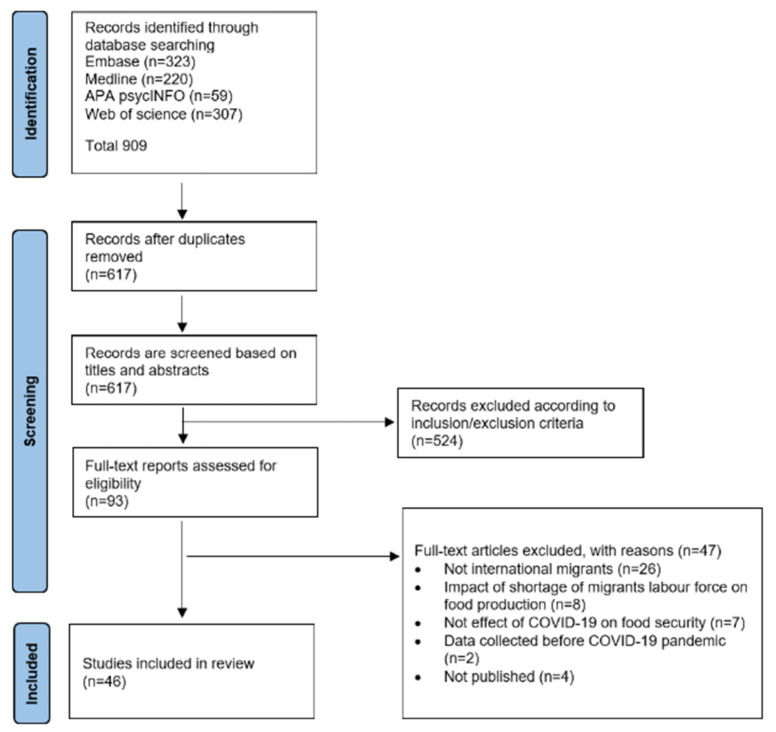
PRISMA flowchart.

**Figure 3 ijerph-20-05273-f003:**
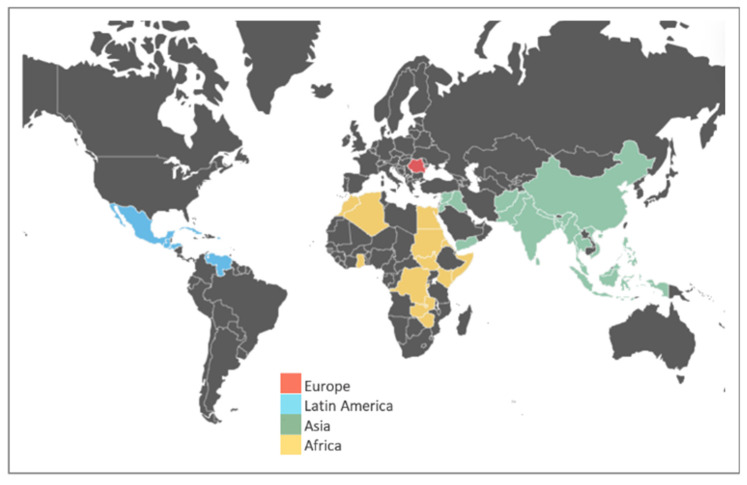
Country of origin of the migrant samples of the studies selected (Marshall Islands in Oceania not visible on the map).

**Table 1 ijerph-20-05273-t001:** PICO/PECO strategy.

PICO/PECO Strategy		Search Terms
P	Participants	International migrants
I or E	Intervention/exposure	COVID-19
C	Control	Not applied for this study
O	Outcome	Food (in)security

**Table 2 ijerph-20-05273-t002:** Description of the 46 studies included.

Variable	Total-N
Total-N	46
Country in which study was conducted	
America	23
USA	19
Canada	2
Trinidad and Tobago	1
Peru	1
Asia	11
Bangladesh	4
China	1
Gulf Cooperation Council countries	1
Israel	2
Singapore	1
Malaysia	1
Iraq, Lebanon, Turkey, Jordan, and Syria	1
Africa	6
Rwanda	1
South Africa	2
Uganda	2
Ethiopia	1
Europe	4
Switzerland	1
United Kingdom	2
Netherlands	1
Oceania	2
Australia	2
Year published	
2020	9
2021	14
2022	23
Publication type	
Original research	37
Qualitative design	9
Quantitative design	16
Mixed design	12
Commentary	4
Viewpoint	1
Opinion piece	1
Literature search	2
Letter to the editor	1

**Table 3 ijerph-20-05273-t003:** Link between the COVID-19 pandemic and food insecurity among international migrants.

Link between COVID-19 Pandemic and Migrants’ Food (in) Security	Number of Studies
Increased financial hardship (job loss, decreased working hours, decreased income, and elevated food prices)	28
Migrants avoided utilising humanitarian food assistance due to legal residency policies and/or fear of deportation	13
Suspended/reduced humanitarian assistance	7
Lack of safe, high-quality, and/or culturally appropriate food during the pandemic affected migrants’ food consumption	6
Closure of schools that offered meals for children	6
Unavailability of certain food items either due to disturbed supply chain or panic buying	5
Lack of physical access due to lockdown measures	3
Language as a barrier to access governmental programmes and food aid	2
Workers staying at home not able to access food at workplace	1
Worries of gaining weight during home quarantine (reducing physical activity) that led to skipping meals	1

**Table 4 ijerph-20-05273-t004:** Classification of articles by the impact of food insecurity on migrants’ health during the COVID-19 pandemic.

Physical Health (8 of 46 Studies)	Mental Health (18 of 46 Studies)
Digestive problems Reduced immunity Undernourishment Vitamin deficiencies Worsened COVID-19 infection Long-term consequences: impairment of children’s development, increased susceptibility to and morbidity of diseases Increased children’s BMI	Anxiety due to food insecurity Depression due to food insecurity Stress due to lack of access to culturally preferable food Sadness for staying hungry due to skipping poor-quality meals Low self-esteem resulting from utilising food assistance services Fear due to difficulty in obtaining food Worries regarding safety of food during the pandemic Somatisation due to anxiety for food insecurity

## Data Availability

Not applicable.

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
