# Peer review of "Food Insecurity among International Migrants during the COVID-19 Pandemic: A Scoping Review"

_ijerph, 2023, doi:10.3390/ijerph20075273_

Round 1

Reviewer 1 Report

This paper provides a scoping review of the literature that has been published to date, and available among a selected number of databases, on how migrant food insecurity changed or was affected as a result of pandemic conditions. The paper is well-written and straightforward in reporting the findings from the review process and the transparency around methodology is a strength. In light of the current global climate and the pandemic's effects still unraveling, I think it is important that the authors acknowledge that this is only the beginning of research that will be conducted on this particular subject and that similar studies should commence at periodic intervals as we get farther from the start of the pandemic. Research of this type takes significant investment time and resources and thus it is very probable that multiple studies have not yet arrived at a phase of publication. 

Aside from some minor grammatical and syntax changes, such as changing from help organizations to perhaps humanitarian organizations, "cycles" instead of "circles", "food insecure" rather than "unsecure", etc. the authors should contextualize how they developed the pyramid framework for food security/insecurity as it seems informed by other work. 

Author Response

Dear Sir/Madam,

Thank you for considering our manuscript for publication in the International Journal of Environmental Research and Public Health. We are grateful for the opportunity to respond to your comments. See below for responses to the  comments point by point. We believe this has improved the manuscript substantially, and hereby submit the manuscript revised according to the comments.

Comment 1: This paper provides a scoping review of the literature that has been published to date, and available among a selected number of databases, on how migrant food insecurity changed or was affected as a result of pandemic conditions. The paper is well-written and straightforward in reporting the findings from the review process and the transparency around methodology is a strength. In light of the current global climate and the pandemic's effects still unraveling, I think it is important that the authors acknowledge that this is only the beginning of research that will be conducted on this particular subject and that similar studies should commence at periodic intervals as we get farther from the start of the pandemic. Research of this type takes significant investment time and resources and thus it is very probable that multiple studies have not yet arrived at a phase of publication. 

Author response: Thank you so much for your comments. We agree that multiple studies could have been missing due to limiting the search to November 2021. For this reason, the search was updated until November 2022 in the revised manuscript and updated all the information.

Comment 2: Aside from some minor grammatical and syntax changes, such as changing from help organizations to perhaps humanitarian organizations, "cycles" instead of "circles", "food insecure" rather than "unsecure", etc. the authors should contextualize how they developed the pyramid framework for food security/insecurity as it seems informed by other work. 

Author response: All grammatical and syntax changes were included in the revised manuscript. Furthermore, we added how the pyramid framework was developed (lines 50 – 55): “Food security is a complex and multidimensional concept. “To reflect upon this, we used a framework developed by Gribson, M. based on the Committee on Food security and the Food Insecurity and Vulnerability Information and Mapping Systems initiative. This framework divided food security into four different levels (individual, household, national and underlying conditions). There are several factors related to food security in each (Figure 1), including individual health status.”

Reviewer 2 Report

It would be ideal to have a research question to guide the search and even a hypothesis. Within the methods they point out the identification of a research question, but apparently refer to that of the works that were reviewed to be included. What kinds of questions did they consider?

In this same sense, the objective of the study talks about identifying gaps in knowledge, but does not define what gaps exist, since there will hardly be an exploration of the totality of knowledge. This is why it is important to have a better definition of the objective and very useful to have a question.

It is important to review the way in which Covid-19 has affected (epidemiological data) mobile groups. Is there really greater contagion compared to other especially vulnerable groups? Why should they be a relevant group for the authorities? Do the epidemiological surveillance systems in Covid-19 include the different groups of mobile populations?

How is the data for food insecurity in these same groups? This would support the discussion and conclusions where attention is called for governments to consider mobile groups as priorities.

The search period is not clear and it strikes me that it is limited to November 2021 and the article is being submitted at the end of 2022. I suggest updating it until at least November 2022.

Why, if it is recognized that most of the information comes from organizations and not from scientific publications, was the review of those documents not included? It would be good to delve into that discussion about what knowledge is and what are the gaps in knowledge and its legitimacy, as well as the impact on the specific issue of food insecurity and migration.

How do you define migrants in your work? Obviously, a migrant in transit without immigration documents is not the same as an international student or a person seeking asylum or refuge. These differences are substantive.

I suggest not talking about migrants with legal or illegal status and if it is maintained then to make explicit in which country this connotation is given to migrants.

It would be very interesting to know the level of affectation that mobile groups have with respect to food insecurity (mild, moderate or severe) and how it is measured in the different studies. In the same way, it would be good to know the coping strategies used and the impact they have on health. Are these issues reported in studies?

Residents and minors in schools are considered in the discussion. It is not clear to me why they are included when they are not the target groups of this review. I think this problem is a consequence of the lack of definition of work when talking about migrants in general. The same happens when explaining the issue of work and the income of migrants.  It is not clear how a migrant in transit compared to an asylum seeker can be affected in this regard. You have to specify what type of mobile group they refer to throughout the text. The same for the issues of changes in diet, isolation during quarantine, among others.

The limitations of the work are much greater than those that arise and have to do with the lack of research question, definition of migrant and other aspects that I have mentioned. Much more thought should be given to the points at issue here in order to better resolve the work. For me it remains at a very descriptive and general level which dilutes its relevance.

Author Response

Dear Sir/Madam,

Thank you for considering our manuscript for publication in the International Journal of Environmental Research and Public Health. We are grateful for the opportunity to respond to your comments. See below for responses to the comments point by point. We believe this has improved the manuscript substantially, and hereby submit the manuscript revised according to the comments.

Comment 1: It would be ideal to have a research question to guide the search and even a hypothesis. Within the methods they point out the identification of a research question, but apparently refer to that of the works that were reviewed to be included. What kinds of questions did they consider?

Author response:  We added the research question in the method section of the manuscript (lines 97 – 99): “the following research question was identified: what kind of peer-reviewed literature has been published about food security among international migrants during the COVID-19 pandemic?”

Furthermore, the introduction has been expanded to explain the reason for us to elaborate the research question:

Lines 69 – 83:

Food security has obtained increasing attention during the COVID-19 pandemic, with a growing number of papers describing the situation for the majority population. These studies acknowledge the complexity of the associations between the pandemic and the key role of sociodemographic factors. However, they seldom include migration as an additional factor to consider in these associations. So far, most of the information regarding food security/insecurity among migrants during the pandemic comes from NGOs and international humanitarian organisations working in low- and middle countries. Although this information is extremely valuable, humanitarian organizations often have advocacy as one of their roles and do not have the resources to conduct independent investigations available to the research community.

Reliable peer-reviewed evidence of high quality about food insecurity among migrants is necessary for researchers to understand the intricacies of associations between health and migration background and for policymakers to make adequate decisions, specifically during health crises like the pandemic, when health, economy and health services are compromised.”

Lines 86 – 92: “In the light of this information, we will try to identify any gap in knowledge related to the levels and factors of the food insecurity framework, with a special emphasis on health, as well as to different risk factors related to migration background. Even though food insecurity is of utter importance for internally displaced people also, given that the pandemic posed strict constraints to movements across national borders, for this paper, we study migrants that have moved to a new country.”

Comment 2: In this same sense, the objective of the study talks about identifying gaps in knowledge, but does not define what gaps exist, since there will hardly be an exploration of the totality of knowledge. This is why it is important to have a better definition of the objective and very useful to have a question.

Author response:  We agree with the reviewer that the objective is unclear. We have updated this information in the introduction, methods and discussion sections.

First, we added the research question in methods (lines 97 - 99): “First, the following research question was identified: what kind of peer-reviewed literature has been published about food security among international migrants during the COVID-19 pandemic?

In addition, we have added more information in the introduction section (lines 86 – 89): “In the light of this information, we will try to identify any gap in knowledge related to the levels and factors of the food insecurity framework, with a special emphasis on health, as well as to different risk factors related to migration background.”

Finally, in the discussion section, we added the following reflection based on the results of our scoping (lines 343 – 350): “Last, by setting our results in the light of a food security framework and the recommendations for research in the research field of migrant health, we have suggested some gaps in the literature in terms of volume of literature, migrant-related factors covered, description and instrumentalization of food insecurity as the main outcome or explaining factor. However, a scoping review gives some indications of the research published but does not cover all literature. Nevertheless, we acknowledge our scoping review's limitations in detecting all literature and that other ways of looking for gaps might be more relevant.”  

Comment 3: It is important to review the way in which Covid-19 has affected (epidemiological data) mobile groups. Is there really greater contagion compared to other especially vulnerable groups? Why should they be a relevant group for the authorities? Do the epidemiological surveillance systems in Covid-19 include the different groups of mobile populations? How is the data for food insecurity in these same groups? This would support the discussion and conclusions where attention is called for governments to consider mobile groups as priorities.

Author response: We agree that our paper could be read as if we were talking about other groups in vulnerable situations apart from migrants and have therefore changed some words in the first part of the introduction (lines 32 - 39) “Indeed, the pandemic has disproportionately impacted refugees, asylum seekers, and other migrants who live in vulnerable conditions and have a higher risk of being infected and dying from COVID-19. However, migrants constitute a heterogenic group that has been affected differently depending on factors such as their country of origin and destination, the reason for migration, and their status of documentation, among others. In addition, people in vulnerable situations experienced more financial hardship and other social consequences during the pandemic, which might be attributed to pre-existent socioeconomic inequalities.”

Data on migrants and the pandemics worldwide shows an overrepresentation among the infected and the dead. Unfortunately, not much information is available for other groups in vulnerable populations, and in addition other groups apart for international migrants were out of the scope of this paper. Furthermore, the reviewer points out an enormous challenge in this field of research that we also highlight in our discussion:

Lines 324 – 333: “In regard to the recommendations in the field of research on migration and health, although definitions and classifications of migrant groups were incomplete or omitted in some of the studies selected, most of the migrants presented in the studies included in this scoping review belong to groups in very vulnerable situations: refugees and asylum seekers, undocumented migrants, and precarious migrant workers. Based on the existing evidence, different migrant workers, not only those in precarious situations, have experienced the loss of jobs and/or decreasing wages affecting their health and living conditions. More focus is needed on the impact COVID-19 has had on the health and food security status of different types of international migrant workers in different countries.

Lines 343 – 346: “Last, by setting our results in the light of a food security framework and the recommendations for research in the research field of migrant health, we have suggested some gaps in the literature in terms of volume of literature, migrant-related factors covered.”

Comment 4: The search period is not clear and it strikes me that it is limited to November 2021 and the article is being submitted at the end of 2022. I suggest updating it until at least November 2022.

Author response:  We agree that multiple studies could have been missing due to limiting the search to November 2021. For this reason, the search was updated until November 2022 in the revised manuscript. We clarified the search period in lines 120 – 121: “We limited the search until November 2022. The final search was carried out on 26 January 2023.”

Comment 5: Why, if it is recognized that most of the information comes from organizations and not from scientific publications, was the review of those documents not included? It would be good to delve into that discussion about what knowledge is and what are the gaps in knowledge and its legitimacy, as well as the impact on the specific issue of food insecurity and migration.

Author response:  We did not include reports from organizations as we wanted to identify if the academic research community has been focused on investigating this topic. This is reflected in the main research question, which specifies in line 98 “what kind of peer-reviewed literature”.

The challenge with NGOs is that their role is often one of advocacy, and not always have the resources to conduct independent research projects. We have added this information in the introduction (lines 76 – 83): “Although this information is extremely valuable, humanitarian organizations often have advocacy as one of their roles and do not have the resources to conduct independent investigations available to the research community.

Reliable peer-reviewed evidence of high quality about food insecurity among migrants is necessary for researchers to understand the intricacies of associations between health and migration background and for policymakers to make adequate decisions, specifically during health crises like the pandemic, when health, economy and health services are compromised.”

Comment 6: How do you define migrants in your work? Obviously, a migrant in transit without immigration documents is not the same as an international student or a person seeking asylum or refuge. These differences are substantive.

Author response:  We agree that the definition of migrant in our work was not clear. We have added information in the introduction to clarify this.

Lines 32 – 36: “the pandemic has disproportionately impacted refugees, asylum seekers, and other migrants who live in vulnerable conditions and have a higher risk of being infected and dying from COVID-19. However, migrants constitute a heterogenic group that has been affected differently depending on factors such as their country of origin and destination, the reason for migration, and their status of documentation, among others.”

Lines 89 – 92: “Even though food insecurity is of utter importance for internally displaced people also, given that the pandemic posed strict constraints to movements across national borders, for this paper, we study migrants that have moved to a new country.”

Comment 7: I suggest not talking about migrants with legal or illegal status and if it is maintained then to make explicit in which country this connotation is given to migrants.

Author response:  We agree with the reviewer that a person cannot be illegal. Terminology in this particular issue is complicated because it changes in the different countries: the terms legal status /documented/undocumented are both used in the literature. In our new manuscript, the term “legal status” has been changed to “documentation status” in lines 18, 144, and 213.

Comment 8: It would be very interesting to know the level of affectation that mobile groups have with respect to food insecurity (mild, moderate or severe) and how it is measured in the different studies. In the same way, it would be good to know the coping strategies used and the impact they have on health. Are these issues reported in studies?

Author response:  We agree with the reviewer that this had been important information, but the level of food insecurity and coping strategies are, unfortunately, seldom reported in the papers. There are probably a number of reasons for this: first, the tools used to measure food security are different, and not all allow specification in levels; second, food security is included as a secondary objective or even considered as an adjusting variable in several of the papers. Reflecting upon the reviewer´s comment, we have now included this gap in knowledge regarding food security (the lack of knowledge of the degree of food security) in the discussion in the paper, as it is of relevance:

Lines 319 – 321: “Last, food security was not always the primary outcome or explaining factor and was measured with different instruments, often non-comparable and poorly described”

Lines 343 – 347:  “Last, by setting our results in the light of a food security framework and the recommendations for research in the research field of migrant health, we have suggested some gaps in the literature in terms of volume of literature, migrant-related factors covered, description and instrumentalization of food insecurity as the main outcome or explaining factor

Comment 9: Residents and minors in schools are considered in the discussion. It is not clear to me why they are included when they are not the target groups of this review. I think this problem is a consequence of the lack of definition of work when talking about migrants in general. The same happens when explaining the issue of work and the income of migrants.  It is not clear how a migrant in transit compared to an asylum seeker can be affected in this regard. You have to specify what type of mobile group they refer to throughout the text. The same for the issues of changes in diet, isolation during quarantine, among others.

Author response:  We agree with the reviewer that this information needs clarification. The results we have identified and reported in our scoping review refer to different migrant groups; to avoid confusions, we have specified “migrant children” (line 279) and “migrants’ psychological disorders” (line 291). Regarding the type of mobile group, it has not been possible to compare migrants groups as many of the papers lack definitions and proper classifications, making our classification work impossible. This is a general problem in the migration and health research field as stated by the UCL-Lancet commission on migration and health. We have highlighted this gap in the discussion section:

Lines 325 - 326:  “definitions and classifications of migrant groups were incomplete or omitted in some of the studies selected

Lines 343 – 347: “Last, by setting our results in the light of a food security framework and the recommendations for research in the research field of migrant health, we have suggested some gaps in the literature in terms of volume of literature, migrant-related factors covered, description and instrumentalization of food insecurity as the main outcome or explaining factor.”

Comment 10: The limitations of the work are much greater than those that arise and have to do with the lack of research question, definition of migrant and other aspects that I have mentioned. Much more thought should be given to the points at issue here in order to better resolve the work. For me it remains at a very descriptive and general level which dilutes its relevance.

Author response: We agree with the reviewer that the limitations are greater than those mentioned in our first manuscript. In this new version of the manuscript, we added additional information in the introduction (see previous responses to Reviewer 2 comments) and based on this we added more limitations. Limitations in this scoping review were defined at two levels and are more clear in the discussion section:

The limitation of the papers we describe as related to the food security framework and the migration and health field. We have called these gaps in the discussion and relate to general matters (definitions, classifications), groups chosen according to country of origin or destination or reason for migration, disparity of instruments measuring food insecurity and its levels, report of food security and link of food insecurity to health levels as expressed in the framework.

Lines 310 – 333: “It is noteworthy that of the several hundred thousand peer-reviewed papers related to COVID-19 published during the pandemic, only a few articles discussed how COVID-19 impacted the food security of international migrants. Furthermore, only four of the studies in this scoping review were carried out in European countries, revealing a lack of research awareness on this subject and a disregard of the fact that food insecurity among migrants is also increasing in high-income countries like Norway. Based on the food security framework used in this study, we have found several gaps in research and knowledge. First, only two studies focused on National/Regional and Context/Underlying conditions levels. Second, few studies connected food security and the health status of migrants. Last, food security was not always the primary outcome or explaining factor and was measured with different instruments, often non-comparable and poorly described. Addressing these gaps would help identify factors at all levels and understand their connections. This is crucial to make comparisons and implement policies and interventions.

In regard to the recommendations in the field of research on migration and health, although definitions and classifications of migrant groups were incomplete or omitted in some of the studies selected, most of the migrants presented in the studies included in this scoping review belong to groups in very vulnerable situations: refugees and asylum seekers, undocumented migrants, and precarious migrant workers. Based on the existing evidence, different migrant workers, not only those in precarious situations, have experienced the loss of jobs and/or decreasing wages affecting their health and living conditions. More focus is needed on the impact COVID-19 has had on the health and food security status of different types of international migrant workers in different countries.”

Lines 343 – 350: “by setting our results in the light of a food security framework and the recommendations for research in the research field of migrant health, we have suggested some gaps in the literature in terms of volume of literature, migrant-related factors covered, description and instrumentalization of food insecurity as the main outcome or explaining factor. However, a scoping review gives some indications of the research published but does not cover all literature. Nevertheless, we acknowledge our scoping review's limitations in detecting all literature and that other ways of looking for gaps might be more relevant.”

On the other side, our scoping review has its own limitations, which are explained in lines 334 – 343: “This scoping review has some limitations. First, because it includes only studies in English, it is possible that many relevant national and international studies in other languages have been missed. Also, papers that do not describe the direct association between the pandemic and food insecurity on migrants themselves were excluded, which led us to ignore valuable information on how COVID-19 disturbed general food supply chains and the food production process which exacerbated food insecurity in general. The papers obtained in our search were heterogeneous in terms of definitions (often using food security and insecurity interchangeably) and design and method, which made comparisons and finding common topics challenging. However, four different databases were used for the search, making our scoping review comprehensive and multidisciplinary.”

Reviewer 3 Report

This review paper draws attention to a critically important issue.  Unfortunately, by the time that the authors applied their inclusion/exclusion criteria, they have only 17 papers left, half of which were published in 2020. Thus, the main message of the paper (an important one) is that there is very little research to date and not much to say on this, which is their primary conclusion.  So, in a way, the paper becomes a review of what isn’t.  Thus, to actually produce a substantive review, there is a sleight of hand as most of the Discussion discusses literature that is outside the scope of the review as defined (i.e. notes 38 to 60).  Although it is an interesting discussion it is not justifiable in terms of the methodology which the paper lays out.  To me, this is a major flaw of the paper and I therefore cannot recommend publication  unless the author(s) do the following before re-submitting:

1.       Included a much larger group of studies by (a) adjusting the exclusion criteria, (b) by a more thorough search (I know of articles published in this time frame that seem to have been missed); (c) by bringing the time frame closer to the present (say end-2022) as a lot more has been published during the last year; and (d) by not limiting the search to published journal articles but including book chapters, reports, working and discussion papers.  A simple Google Scholar search would identify many of these.

2.       There is a fundamental disconnect in the paper between the opening discussion on food security and the rest of the paper.  The figure is interesting but the contents need elaboration (for example, there is nothing about food environments, two of the labels in National/Regional are duplicates, and national/regional need to be separated not lumped together).  Also, the four standard dimensions of food security are not included (let alone the 2 new dimensions of agency and sustainability proposed by the CFS HLPE).  However, the disconnect is that the introductory discussion is not joined up with the contents of the review.  So, for example, it would be more useful to return to this figure in the Conclusion and say how it should be modified from its general character to be appropriate to international migration.

3.       The paper provides an extended discussion of the nature of food security (notes 1-17) but provides no equivalent discussion of international migration.  Thus, the highly varied and differentiated character of international migration is assumed away.  At the very least it would be helpful to develop or add a typology of different forms of migration since the impacts of COVID on migrant food security is likely to vary considerably by type.  This, then, also becomes another way of classifying studies in a review.   

Author Response

Dear Sir/Madam,

Thank you for considering our manuscript for publication in the International Journal of Environmental Research and Public Health. We are grateful for the opportunity to respond to your comments. See below for responses to the comments point by point. We believe this has improved the manuscript substantially, and hereby submit the manuscript revised according to the comments.

Comment 1:  Included a much larger group of studies by (a) adjusting the exclusion criteria, (b) by a more thorough search (I know of articles published in this time frame that seem to have been missed); (c) by bringing the time frame closer to the present (say end-2022) as a lot more has been published during the last year; and (d) by not limiting the search to published journal articles but including book chapters, reports, working and discussion papers.  A simple Google Scholar search would identify many of these.

Author response: We agree with the reviewer that the search should be updated.  For this reason, the search was updated until November 2022 in the revised manuscript and we identified 29 more studies from 2021 and 2022. In addition, we adjusted the exclusion criteria in lines 105 – 107: “We excluded studies on internal migrants or minorities and ethnic groups, systematic and scoping reviews, and articles that discussed the effect of COVID-19 on agriculture and food production due to the shortage of migrant farmworkers.”

Finally, we clarified in the introduction section the reason why we are including only peer-reviewed evidence (lines 73 – 83): “So far, most of the information regarding food security/insecurity among migrants during the pandemic comes from NGOs and international humanitarian organisations working in low- and middle countries. Although this information is extremely valuable, humanitarian organizations often have advocacy as one of their roles and do not have the resources to conduct independent investigations available to the research community.

Reliable peer-reviewed evidence of high quality about food insecurity among migrants is necessary for researchers to understand the intricacies of associations between health and migration background and for policymakers to make adequate decisions, specifically during health crises like the pandemic, when health, economy and health services are compromised.

Comment 2:  There is a fundamental disconnect in the paper between the opening discussion on food security and the rest of the paper.  The figure is interesting but the contents need elaboration (for example, there is nothing about food environments, two of the labels in National/Regional are duplicates, and national/regional need to be separated not lumped together).  Also, the four standard dimensions of food security are not included (let alone the 2 new dimensions of agency and sustainability proposed by the CFS HLPE).  However, the disconnect is that the introductory discussion is not joined up with the contents of the review.  So, for example, it would be more useful to return to this figure in the Conclusion and say how it should be modified from its general character to be appropriate to international migration. 

Author response: We agree with the reviewer comments about the disconnection between the discussion and the rest of the paper, including the framework we are using. We added more information about the framework in the introduction (lines 50 - 55): “To reflect upon this, we used a framework developed by Gribson, M. based on the Committee on Food security and the Food Insecurity and Vulnerability Information and Mapping Systems initiative. This framework divided food security into four different levels (individual, household, national and underlying conditions). There are several factors related to food security in each (Figure 1), including individual health status”.

Furthermore, we have reorganized the discussion section to join it with the contents of the review. All gaps have been moved to the end of the discussion section and we have included several extra gaps and limitations connected to the framework and the migration and health field:

Lines 315 – 333: “Based on the food security framework used in this study, we have found several gaps in research and knowledge. First, only two studies focused on National/Regional and Context/Underlying conditions levels. Second, few studies connected food security and the health status of migrants. Last, food security was not always the primary outcome or explaining factor and was measured with different instruments, often non-comparable and poorly described. Addressing these gaps would help identify factors at all levels and understand their connections. This is crucial to make comparisons and implement policies and interventions.

In regard to the recommendations in the field of research on migration and health, although definitions and classifications of migrant groups were incomplete or omitted in some of the studies selected, most of the migrants presented in the studies included in this scoping review belong to groups in very vulnerable situations: refugees and asylum seekers, undocumented migrants, and precarious migrant workers. Based on the existing evidence, different migrant workers, not only those in precarious situations, have experienced the loss of jobs and/or decreasing wages affecting their health and living conditions. More focus is needed on the impact that COVID-19 has had on the health and food security status of different types of international migrant workers in different countries.”

Lines 343 – 350: “by setting our results in the light of a food security framework and the recommendations for research in the research field of migrant health, we have suggested some gaps in the literature in terms of volume of literature, migrant-related factors covered, description and instrumentalization of food insecurity as the main outcome or explaining factor. However, a scoping review gives some indications of the research published but does not cover all literature. Nevertheless, we acknowledge our scoping review's limitations in detecting all literature and that other ways of looking for gaps might be more relevant.”

Comment 3:  The paper provides an extended discussion of the nature of food security (notes 1-17) but provides no equivalent discussion of international migration.  Thus, the highly varied and differentiated character of international migration is assumed away.  At the very least it would be helpful to develop or add a typology of different forms of migration since the impacts of COVID on migrant food security is likely to vary considerably by type.  This, then, also becomes another way of classifying studies in a review.   

Author response: We agree with the reviewer that we missed discussing the highly varied and differentiated character of international migrants and how the impact of COVID on migrant food security can vary considering the type of migrant. We have updated the new manuscript to include this information:

Lines 32 to 39: “Indeed, the pandemic has disproportionately impacted refugees, asylum seekers, and other migrants who live in vulnerable conditions and have a higher risk of being infected and dying from COVID-19. However, migrants constitute a heterogenic group that has been affected differently depending on factors such as their country of origin and destination, the reason for migration, and their status of documentation, among others. In addition, people in vulnerable situations experienced more financial hardship and other social consequences during the pandemic, which might be attributed to pre-existent socioeconomic inequalities.”

Lines 89 to 92: “Even though food insecurity is of utter importance for internally displaced people also, given that the pandemic posed strict constraints to movements across national borders, for this paper, we study migrants that have moved to a new country.”

We also updated the discussion section to emphasize that definitions and classifications of migrant groups were incomplete or omitted in some of the studies selected and more focus is needed on food security status of different types of international migrants (lines 324 to 333): “In regard to the recommendations in the field of research on migration and health, although definitions and classifications of migrant groups were incomplete or omitted in some of the studies selected, most of the migrants presented in the studies included in this scoping review belong to groups in very vulnerable situations: refugees and asylum seekers, undocumented migrants, and precarious migrant workers. Based on the existing evidence, different migrant workers, not only those in precarious situations, have experienced the loss of jobs and/or decreasing wages affecting their health and living conditions. More focus is needed on the impact COVID-19 has had on the health and food security status of different types of international migrant workers in different countries.”

Reviewer 4 Report

This paper discusses various dimensions of food security among the international migrants (mostly vulnerable ones) during the COVID-19 pandemic. It is a study primarily conducted on secondary resources. The study itself is well-done, but the main problem with the text is that it lacks in two prominent sections: the theoretical framing, and the review of existing literature that has studied international migrants and food security issues in the COVID-19 period. For a study like this, even if the theoretical background can be sidelined (because of the empirical and practical nature of the study), the reference to existing literature and a literature review are a must, so I would request the author add a 500–1000 word literature review of existing studies on food security, COVID-19, and international migrants to this study.

Author Response

Dear Sir/Madam,

Thank you for considering our manuscript for publication in the International Journal of Environmental Research and Public Health. We are grateful for the opportunity to respond to your comments. See below for responses to the comments point by point. We believe this has improved the manuscript substantially, and hereby submit the manuscript revised according to the comments.

Comment 1:  This paper discusses various dimensions of food security among the international migrants (mostly vulnerable ones) during the COVID-19 pandemic. It is a study primarily conducted on secondary resources. The study itself is well-done, but the main problem with the text is that it lacks in two prominent sections: the theoretical framing, and the review of existing literature that has studied international migrants and food security issues in the COVID-19 period. For a study like this, even if the theoretical background can be sidelined (because of the empirical and practical nature of the study), the reference to existing literature and a literature review are a must, so I would request the author add a 500–1000 word literature review of existing studies on food security, COVID-19, and international migrants to this study.

Author response: We agree with the reviewer that the theoretical background should be updated. In our new manuscript, we have included more information and references regarding international migrants, COVID-19 and migrants, and food security and the pandemic for the general population.  

Lines 32 – 41: “Indeed, the pandemic has disproportionately impacted refugees, asylum seekers, and other migrants who live in vulnerable conditions and have a higher risk of being infected and dying from COVID-19. However, migrants constitute a heterogenic group that has been affected differently depending on factors such as their country of origin and destination, the reason for migration, and their status of documentation, among others. In addition, people in vulnerable situations experienced more financial hardship and other social consequences during the pandemic, which might be attributed to pre-existent socioeconomic inequalities. Poorer socioeconomic conditions can in turn increase vulnerability to infection and disease, creating vicious cycles that are reinforced during a pandemic.”

Lines 69 – 78: “Food security has obtained increasing attention during the COVID-19 pandemic, with a growing number of papers describing the situation for the majority population. These studies acknowledge the complexity of the associations between the pandemic and the key role of sociodemographic factors. However, they seldom include migration as an additional factor to consider in these associations. So far, most of the information regarding food security/insecurity among migrants during the pandemic comes from NGOs and international humanitarian organisations working in low- and middle countries. Although this information is extremely valuable, humanitarian organizations often have advocacy as one of their roles and do not have the resources to conduct independent investigations available to the research community.”

Lines 89 – 92: “Even though food insecurity is of utter importance for internally displaced people also, given that the pandemic posed strict constraints to movements across national borders, for this paper, we study migrants that have moved to a new country.”

Round 2

Reviewer 2 Report

The authors have made a great job responding to previous comments and I think the paper is solid and can be published.

Reviewer 4 Report

The authors have taken all of the comments into account and significantly improved the paper. It can be published now.